# Affordable Generative Agents

**Yangbin Yu**                                                                    *yangbinyu@tencent.com*
*Tencent*

**Qin Zhang**                                                                     *adrienzhang@tencent.com*
*Tencent*

**Junyou Li**                                                                     *junyouli@tencent.com*
*Tencent*

**Qiang Fu**                                                                      *leonfu@tencent.com*
*Tencent*

**Deheng Ye**                                                                     *dericye@tencent.com*
*Tencent*

**Reviewed on OpenReview:** *https://openreview.net/forum?id=7tlYbcq5DY*

## Abstract

The emergence of large language models (LLMs) has significantly advanced the simulation of believable interactive agents. However, the substantial cost on maintaining the prolonged agent interactions poses challenge over the deployment of believable LLM-based agents. Therefore, in this paper, we develop Affordable Generative Agents (AGA), a framework for enabling the generation of believable and low-cost interactions on both agent-environment and inter-agents levels. Specifically, for agent-environment interactions, we substitute repetitive LLM inferences with learned policies; while for inter-agent interactions, we model the social relationships between agents and compress auxiliary dialogue information. Extensive experiments on multiple environments show the effectiveness and efficiency of our proposed framework. Also, we delve into the mechanisms of emergent believable behaviors lying in LLM agents, demonstrating that agents can only generate finite behaviors in fixed environments, based upon which, we understand ways to facilitate emergent interaction behaviors. Our code is publicly available at: `https://github.com/AffordableGenerativeAgents/Affordable-Generative-Agents`.

## 1 Introduction

Intelligent agents achieving human-level performance in complex tasks has attracted research efforts for years Mnih et al. (2015); Ye et al. (2020a;b); Zhao et al. (2023). Recently, constructing an agent based on large language models (LLMs) for the simulation of believable human behavior has shown substantial promise (da Rocha Costa, 2019; Park et al., 2022; Lan et al., 2023; Xi et al., 2023a). A believable LLM agent is characterized by its ability to interact with other agents or humans, respond to environmental changes, and generate responses and behaviors that are perceived by humans as authentic, natural, and in alignment with expectations. Such agents could be utilized for diverse applications, such as simulating the potential impacts of policies (Zhang et al., 2023a; Törnberg et al., 2023) or sociological theories on humans (Gao et al., 2023), creating Non-Player Characters (NPCs) (Laird & VanLent, 2001; Gong et al., 2023) in games, and

developing embodied intelligence akin to social robots (Cherakara et al., 2023) or domestic service robots (Puig et al., 2020; Yang et al., 2023b).

However, while we appreciate the potential and capabilities of these agents, their cost implications also merit attention. The crux of the matter is that frequent and prolonged use of LLMs to handle various types of agent interactions can result in substantial cost consumption. Previous studies have highlighted this issue. For instance, Park et al. (2023) construct a virtual town, aka the "Stanford Town", to simulate agents spontaneously exploring and interacting with the environment to produce believable behaviors in an open world, with the entire study incurring costs of thousands of dollars. Wang et al. (2023b) mimics embodied agents, fulfilling both emotional and rational needs through interaction, with simulations of 2 to 3 agents costing several dollars per hour. The cost of believable agent interaction has hindered its application in scenarios requiring large-scale usage. For example, simulating long-term, large-scale policy impacts and sociological phenomena, as well as using LLM agents as NPCs in games.

Many methods have been developed to reduce the cost of invoking LLMs, but most of the research has focused on fixed datasets, such as COQA (Reddy et al., 2019), QQP (Wang et al., 2017), and RTE (Poliak, 2020) datasets. These tasks have clear answers, well-defined difficulty levels, and independence between tasks, typically solvable through single-turn QA interactions. However, the situation is quite different for believable agents, which lack clear answers and involve interactions that require the agent's long-term memory and coordination between different modules. These characteristics make it challenging to apply most existing methods in this context.

Therefore, in this paper, we focus on the believable behavior interaction simulation of LLM agents in open-world scenarios and propose a low-cost framework, termed as *Affordable Generative Agents*. We abstract the behaviors of believable agents into two modes: agent-environment interactions and inter-agent interactions. For the agent-environment interactions, we propose the *Lifestyle Policy* to reduce the redundant costs of generating repeated responses by agents; On the other hand, for the inter-agent interactions, we propose the *Social Memory* to reduce the generation of repeated information in multi-round interactions. To examine the applicability of our techniques, we have conducted extensive experiments using well-known environments, including the "Stanford Town" (Park et al., 2023) and the VirtualHome (Puig et al., 2018), to demonstrate that, while achieving the same performance, the consumption of generating believable agent behaviors can be significantly reduced.

Furthermore, to help understand why our approach works, we delve into the mechanics of how LLM agents generate believable behavior, demonstrating that there is an upper limit to the believable behavior that emerges from LLM agents in a fixed environment. This implies that all agent-environment interactions can be entirely covered by policies. Based upon this finding, we also propose a set of optimization strategies for generating richer believable behavior.

In summary, our contributions are as follows:

- We propose Affordable Generative Agents, a low-cost simulation framework for the believable behavior simulation of LLM agents in open-world scenarios, with optimization strategies for both agent-environment and inter-agent interactions.

- We propose several evaluation methods and conduct extensive experiments in benchmarking environments to validate the effectiveness of our framework.

- We analyze the mechanism of believable behavior generation by LLM agents, demonstrate the upper limit of the richness of emergent behaviors, and further propose corresponding optimization strategies.

## 2 Related Work

This work explores the development of efficient generative behavior leveraging LLM. We discuss the most related works below.

## 2.1 LLM Agents

The development of LLM agents has sparked a myriad of applications. Yang et al. (2023a) leverage multi-turn dialogues with agents to autonomously address NLP tasks. Li et al. (2024) find that the performance of LLMs scales with the number of agents instantiated across a range of tasks. Deng et al. (2023), Nakano et al. (2021), and Yao et al. (2022a) integrate pre-defined APIs with LLMs to enable agents to navigate web pages effectively. Xu et al. (2023) and Light et al. (2023) deploy LLM agents in text game environments, showing the capacity to engage in complex, strategic gameplay. The most prominent among these works is the construction of LLM agents that interact in open environments to create believable behavior. For instance, Park et al. (2023) builds a virtual town simulating the social life of 25 agents, while Brohan et al. (2023) utilizes an LLM to create a robot capable of completing household chores through natural language interaction. Wang et al. (2023a) implements an agent capable of playing Minecraft expertly, when compared to traditional game-playing methods Kanervisto et al. (2022); Lin et al. (2022). Efforts have been made to optimize various modules such as reasoning (Wei et al., 2022; Wang et al., 2022; Xi et al., 2023b), planning (Song et al., 2023; Yao et al., 2022b), reflection (Shinn et al., 2023), and memory (Zhong et al., 2023; Huang et al., 2023b), or to create more realistic interaction environments (Hong et al., 2023b; Huang et al., 2023a). However, the development of affordable and believable agent interactions is missing in the literature. In this paper, we focus on achieving believable agents with the same performance at low cost.

## 2.2 The Cost of LLM Interaction

The cost of using LLM is directly proportional to the number of LLM invocations, the length of the prompt, and the model's performance. The cost issue has always been a key focus in the LLM-related field. Numerous methods have been proposed to reduce the cost of using LLM, such as Cascaded Invocation of LLM (Yue et al., 2023; Chen et al., 2023a; Nottingham et al., 2023), summary (Laban et al., 2023; Arefeen et al., 2023; Mu et al., 2023), and batch (Lin et al., 2023). These methods primarily optimize the cost of single invocation of LLM for QA tasks. Zhang et al. (2023b) has built a cost-efficient multi-agent framework, but it only solves QA tasks on fixed datasets. The simulation of believable behavior in the open world requires uninterrupted calls to the LLM for interaction. Moreover, multiple modules need to combine a large amount of contextual information for LLM reasoning and planning, making cost a more prominent issue. Simply optimizing for single QA invocations cannot effectively reduce costs. Furthermore, the open world does not have a fixed evaluation method, and the difficulty of tasks is hard to categorize, rendering many existing methods unsuitable (Chen et al., 2023a; Zhang et al., 2023b). Kaiya et al. (2023) proposes generative agents with low-cost interactions, but their effectiveness is highly dependent on the tailored scenarios. Hence, there is a pressing need for a low-cost solution that can effectively handle the challenges of believable interaction simulation in various open environments using LLMs. This paper aims to address this gap by proposing a cost-efficient scheme for simulating believable behavior with LLMs in open environments.

## 3 Method

Our work is inspired by Park et al. (2023), who proposed Generative Agents, a framework for simulating believable behaviors of LLM agents in a virtual town. This framework enables agents to perceive their environment and maintain a memory stream to record past interactions and self-reflections. By retrieving relevant content from this memory stream, the framework helps agents generate believable responses. Inspired by this, We achieve generative agents in a low-cost way by optimizing agent-environment interactions and inter-agent interactions. Figure 1 illustrates our method in these two types of interactions, with gray arrows representing the Generative Agents framework's implementation and blue arrows representing our approach. In the following sub-sections, we will describe the optimization strategies for these two categories of interactions in detail.

## 3.1 Agent-environment interactions

The interaction between an agent and its environment can be modeled as a policy function mapping observations to actions. This policy can be implemented through reinforcement learning (Kaelbling et al., 1996)

Figure 1: Our proposed method for optimizing (a) Agent-environment interactions and (b) Inter-agent interactions. The gray arrows represent the baseline (Generative Agents) implementation, while the blue arrows represent our approach.

or rule-based finite state machines (Lee & Yannakakis, 1996), behavior trees (Colledanchise & Ögren, 2018), etc. LLMs trained on a vast amount of data show immense potential in policy formulation (Sun et al., 2023). With carefully designed prompts, LLMs can generate logically consistent and believable decisions. The decision-making capabilities of LLMs have been applied to a wide variety of applications, but the cost associated with each interaction has been overlooked. On the other hand, behavior trees operate without additional costs, but they require pre-written scripts and can not adapt dynamically to environment changes.

Drawing inspiration from the Case-Based Reasoning (CBR) method (Slade, 1991; Spalzzi, 2001), which addresses new problems by adapting solutions to previously solved, similar issues, we introduce a new approach – Lifestyle Policy, which aims to retain the flexible interaction capabilities of the LLMs while achieving low-cost deployment. The functionality of Lifestyle Policy is to minimize costs by reusing the similar inference processes of an LLM. To this end, we devise agent-environment interactions containing two stages: Plan Decomposition and Policy Reuse, expanded as follows.

**Plan Decomposition** The stage occurs when the agent encounters situations it has never seen before. Due to their inherent zero-shot capabilities (Kojima et al., 2022), LLM agents can respond to unseen tasks in a believable way. Given the description of the environment and agent's information, a rough plan or task is generated by an LLM. The Plan Decomposition breaks the plan down into sub-plans and then converts sub-plans into specific actions, just like other LLM multi-agent frameworks (Chen et al., 2023b; Li et al., 2023; Hong et al., 2023a). An additional step involves generating an executable condition based on the action sequence. The determination of execution condition is highly correlated with the environment. For instance, in the case of a domestic robot (Brohan et al., 2023), the condition of a task include the existence of the related interactive items and the consistency of their states. If the plan can be successfully executed, then the whole graph from plan to actions and corresponding condition will be updated to Lifestyle Policy.

**Policy Reuse** The stage occurs when the agent encounters previously seen situations. Given a plan, we use embedding models to extract features from plan descriptions and compare them with stored plan features in the Lifestyle Policy using cosine similarity. When the cosine similarity exceeds a set threshold, we consider it a match for the same plan. Setting the threshold too low may result in mismatched plans, causing agents to perform inappropriate actions, while setting it too high may lead to redundant plans being recorded in

the policy. In our implementation, we set the threshold at 0.97. More details and examples of different similarity levels are provided in the Appendix A. Subsequently, we assess whether the current environment meets the execution condition associated with the retrieved plan by comparing if the interactive items in the current environment exist and have consistent states as specified in the condition. If the condition is met, we proceed with the actions specified in that plan. It is important to note that a single plan may have multiple conditions tailored for different scenarios. We iterate through all the conditions until we find one that the current environment satisfies.

Figure 1(a) illustrates the complete interaction process. Upon receiving an observation that includes environmental data and the agent's personal profile, the LLM agent formulates a plan. For the baseline, all plans will invoke the LLM to decompose and generate corresponding actions. In our method, the Lifestyle Policy retrieves a plan with a similar description and condition met by current environment. If a match is found, the agent executes the corresponding actions. Otherwise, the LLM is invoked to decompose the plan. Due to the superiority of LLMs, the plan decomposition ensures that the actions generated are of high quality, believable, and self-consistent. Meanwhile, the Lifestyle Policy aims to reuse existing plans to avoid unnecessary costs. Due to the high similarity threshold, the replacement policy does not affect the agent's original performance. For instance, the agent Klaus in virtual town generates an activity like *"Klaus is waking up and completing his morning routine."* In the Lifestyle Policy, this activity matches a similar one, *"Klaus is waking up and getting ready for the day,"* and executes the corresponding sub-activities such as *"stretching and getting out of bed," "washing his face and brushing his teeth,"* and *"heading out the door to go to the library."*

In the lifelong simulation of LLM agents, there are a large number of redundant, repetitive, and trivial decisions, such as in Generative Agents, where each agent repeatedly makes decisions like brushing its teeth, washing its face, etc. Replacing the process of invoking LLM reasoning with corresponding policies can save significant cost, which will be shown in our experiments.

## 3.2 Inter-agent interactions

In agent interactions, dialogue is the primary communication method. These agents must integrate substantial auxiliary information into the dialogue prompt to demonstrate advanced cognitive capabilities, distinguishing them from basic chatbots driven by LLMs. For instance, agents need to incorporate relevant information about the dialogue participants and topic-related events retrieved from memory into the dialogue prompt to enrich the conversation. However, this data is often less comprehensive and more fragmented than the knowledge bases of question-answering systems (Cui et al., 2023). Merely appending memory snippets to prompts could reduce response quality and lead to higher processing costs from increased prompt length (Krishna et al., 2022). Furthermore, agent interactions differ depending on their relationships and mutual perceptions in social simulations. Thus, agents need to gather enough information to form contextually appropriate responses without suffering performance drops or incurring extra costs from excessively long prompts.

The gray arrows in Figure 1(b) illustrate the inter-agent interactions provided by the baseline method. Specifically, agents retrieve all relevant events from the memory stream based on the previous round of conversation. LLMs then use these relevant events along with the prior conversation as a prompt to generate the next round of dialogue. However, this approach has several issues. First, trivial and repetitive relevant events can make the prompt excessively long, sometimes exceeding the LLMs' token limit. This not only increases computational costs, especially in multi-agent, multi-turn dialogue scenarios, but also makes it difficult for the LLMs to capture key information, leading to inappropriate responses.

Drawing inspiration from existing text compression technologies (Liu et al., 2023) and social model (McCoy et al., 2011), we introduce the Social Memory module, aimed at reducing the cost of inter-agent interactions and enabling the agents to produce more appropriate dialogues. It consists of two key components: Summary Events and Relationship & Feeling. Summary Events store summaries of relevant events for each interlocutor, while Relationship & Feeling models the social connections between the agent and the interlocutor using keywords such as *"Relationship: acquaintances, Feeling: friendly."* In our implementation, we first retrieve relevant events based on previous dialogue content, filter out new events, and summarize them into Summary

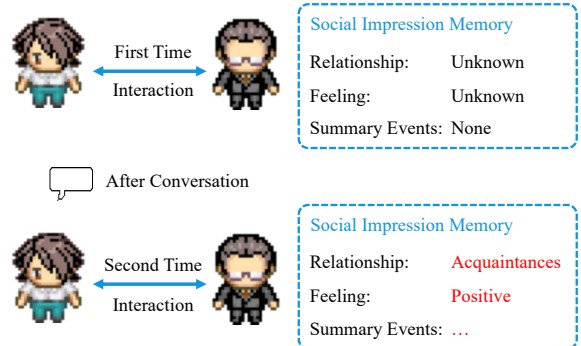

Figure 2: An example of relational evolution driven by the updating of Social Memory following conversational interactions.

Events. These summaries increase information density while controlling text length. We then combine the agent's personal information, event summaries, relationships, feelings, and previous dialogues into a prompt for LLMs to generate the next round of dialogue. After the dialogue ends, we input the current agent's relationship, feelings and the summary of multiple rounds of dialogue into the LLM to update the Relationship & Feeling, as shown in Figure 2. Implementation details of each module of Social Memory are provided in the Appendix B.

Social Memory reduces costs of conversation by compressing auxiliary information, offering precise descriptions of social information to assist agents in responding consistently and appropriately. In the baseline, each conversation requires retrieving 30 to 45 relevant events, which occupy about 2000 tokens in the prompt. In contrast, our method condenses this information to approximately 100 tokens by using Summary Events. It also supplies high-information events to aid agents in generating accurate replies relevant to the conversation. Furthermore, as illustrated in Figure 2, Social Memory updates after each interaction, providing an explicit depiction of interlocutor relationships. For example, we observe that the student Klaus enjoys discussing his studies with others and occasionally mistakes the cafe owner Isabella for a fellow student in baseline. However, with our method, which employs a clear relationship update mechanism, the agent can discern social relationships during conversations. Meanwhile, This mechanism can be employed to investigate the dynamics of relationships between agents, thereby analyzing the evolution of community relations and the emergence of social behaviors.

## 4 Experimental Setup

This section focuses on the experimental environment and evaluation protocol.

### 4.1 Environment

We evaluate our approach under the following environments:

- **Generative Agents**. Park et al. (2023) creates a simulated town, populating it with multiple agents powered by LLMs that interact in natural language, generating believable individual and emergent social behaviors. Agents are required to create daily schedules, which are refined into hourly plans, and then translated into minute-level activities. All interactions are executed through natural language descriptions, accompanied by corresponding sprites and emojis for representation.

  The Generative Agents framework comprises five modules: *Perceive*, *Plan*, *Retrieve*, *Reflect*, and *Act*. The Perceive module is responsible for sensing surrounding game objects. The Plan module generates plans and activities. Retrieve is tasked with retrieving relevant events from memory. Reflect deeply considers past events, and Act determines the location and objects of interaction, as well as generating corresponding emojis.

We implement AGA on Generative Agents framework and conduct experiments in the provided 3-person and 25-person environments.

- **VirtualHome**. Puig et al. (2018) provides a 3D simulated domestic setting where agents engage with their surroundings via programmatic atomic actions. The environment encompasses 115 specific types of items, such as *plate*, *desk*, and *microwave*. Each item possesses attributes that determine the permitted interactions, including *GRABBABLE*, *LIEABLE*, etc., as well as current states like *OPEN* and *CLOSED*. Compared to Generative Agents, VirtualHome offers more concrete and realistic interaction. An interaction involves specific actions, items and the item states.

  We implemented an LLM framework on VirtualHome and applied AGA to it. The detailed implementation is presented in the Appendix E.

## 4.2 Language Models Adopted

For a fair comparison, all agents are empowered with GPT-3.5-Turbo (Wu et al., 2023), same as the Generative Agents work (Park et al., 2023). Besides, we also conduct tests using the more advanced GPT-4 (Achiam et al., 2023).

## 4.3 Evaluation Protocol

We focus on LLM cost and performance. For LLM cost, to avoid regional and time-based price differences, we use the same model to compare the number of tokens consumed by different methods for the same task. To comprehensively evaluate the agent's behavior, we first reuse the evaluation methods of Generative Agents excluding manual assessments. These methods encompass interviewing agents with manually designed questions and conducting case study-specific activities.

Moreover, we attempt to evaluate the emergent social behaviors of generative agents from two other perspectives

- **Relationship Evolution**. The relationships among agents are updated by Social Memory after interactions. The evolution of these relationships signifies the formation of community ties, which can be utilized to examine the social behaviors.

- **Behavioral Assessment**. Chiang & Lee (2023) have validated LLM as an evaluator, demonstrating comparable performance to expert evaluators. Consequently, we employ GPT-4 to evaluate the activities and dialogues generated by LLM agents. A 5-point Likert scale questionnaire is deployed to differentiate whether these activities and dialogues are from human or AI-generated.

For VirtualHome, agents are designed to experience a full day at home, generating household tasks and decomposing them into specific actions. We evaluate the success rate of plan execution.

# 5 Evaluations

We conduct extensive experiments to validate our method, assessing both the token consumption and the performance. Our experiments demonstrate that the AGA framework can reduce the cost while maintaining equivalent performance.

## 5.1 Results on Generative Agents

**Token Consumption** Figure 3 illustrates the ablation study of token consumption based on multiple two game day simulations of 3-person town with GPT-3.5-Turbo. Compared to the baseline, the cost of using only the Lifestyle Policy is 40.2% of the original, while using only the Social Relationship Memory is 58.6%, and the full AGA framework is 31.1%. It should be noted that the advantage of the Lifestyle Policy lies in the reuse of existing LLM-generated strategies. Hence, both Lifestyle Policy and AGA conduct experimental comparisons after running multiple iterations, and then storing a certain number of policies.

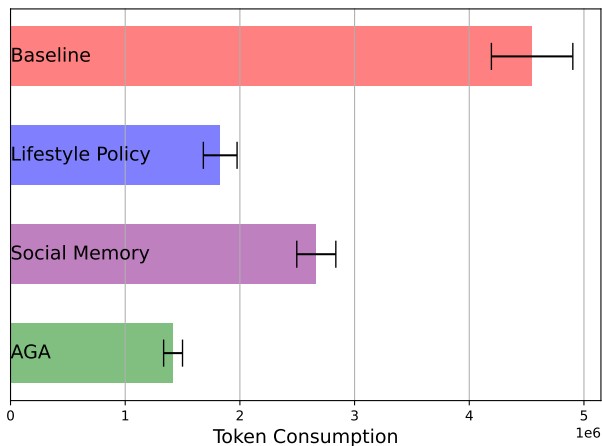

Figure 3: Ablation study on token consumption. **Baseline** means Generative Agents. **Lifestyle Policy** and **Social Memory** mean only using the corresponding module.

Table 1: The token consumption with varying population settings

| Setting | Baseline | AGA | Ratio |
|---------|----------|-----|-------|
| 3 | $4.548M \pm 0.36M$ | $1.417M \pm 0.08M$ | 31.1% |
| 25 | $25.41M \pm 0.96M$ | $10.86M \pm 0.13M$ | 42.7% |

We also present the results of different configuration simulations in Table 1. where the cost for a 25-person town is 42.7% of the original. In the 25-person town, the probability of interactions between agents increases with the rise in population density. After triggering a dialogue, an agent adjusts its plan, invoking the reactions module to handle unexpected events. The occurrence of more events leads to more frequent output of reflections for deeper thinking. In order to ensure the flexibility of the LLM, we did not modify the reactions and reflections modules of the Generative Agents, which incurs additional costs for the 25-person version.

Table 2 provides a detailed comparison across various models and token types in a 3-person town setting. The data suggests that GPT-4 tends to consume slightly more tokens due to its tendency to generate more detailed plans and longer responses. Importantly, the majority of token consumption is attributed to input tokens, a reflection of the need to incorporate substantial relevant information, constraints, and one-shot examples into the prompt. Figure 4 presents the accumulated token consumption over a two-game-day simulation in a 3-person town using GPT3.5-Turbo. The AGA demonstrates slower token consumption compared to the Baseline.

**Case Study** We employ the same method used for Generative Agents to evaluate believable agents in this part. We conduct interviews with agents to evaluate their ability to remember past events, strategize for future tasks based on those memories, react appropriately to unexpected situations, and reflect on their previous actions to improve their future performance. We use the same questions as Generative Agents to inquire about five aspects: *self-knowledge*, *memory*, *plans*, *reactions* and *reflections*. The questions and answers are shown in the appendix C. Results indicate that AGA does not impair the original performance of baseline. Agents can recall past experiences and generate believable responses. For instance, the interviewed agent Klaus Mueller accurately answers questions about his occupation, correctly describes other agents (such as Wolfgang Schulz and Kane Martinez), and is aware of ongoing community events, like Sam running for office and Isabella hosting a Valentine's Day party.

Table 2: The token consumption with varying models

| Method | Model | Input tokens | Output tokens | Total tokens |
|---|---|---|---|---|
| Baseline | GPT3.5-Turbo | $4.212M \pm 0.340M$ | $0.313M \pm 0.024M$ | $4.525M \pm 0.364M$ |
| | GPT-4 | $4.682M \pm 0.398M$ | $0.398M \pm 0.029M$ | $5.069M \pm 0.408M$ |
| AGA | GPT3.5-Turbo | $1.329M \pm 0.085M$ | $0.092M \pm 0.006M$ | $1.420M \pm 0.091M$ |
| | GPT-4 | $1.467M \pm 0.129M$ | $0.130M \pm 0.013M$ | $1.597M \pm 0.142M$ |

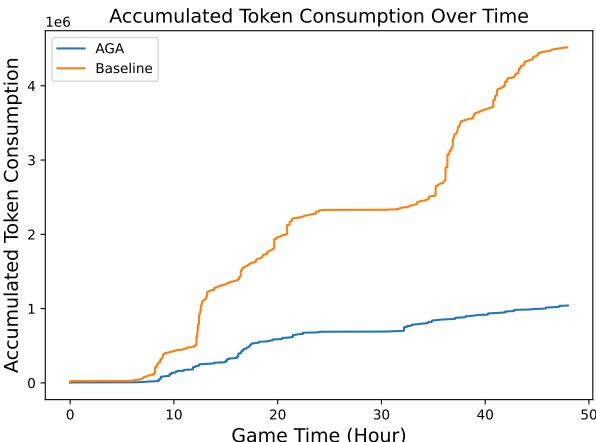

Figure 4: Accumulative token consumption over game time for different methods

In addition, we can validate the emergence of social behavior through specific events. In the 25-person town, Isabella Rodriguez, the owner of the coffee shop, plans to host a Valentine's Day party and invite as many people as possible the following day, while Sam is running for mayor. In the end-to-end evaluation of the baseline, during the two-day simulation, the number of agents aware of Sam's mayoral candidacy is 8, and the number of agents who know about Isabella's party is 13. In our method, after multiple experiments, we observe corresponding numbers ranging from 4 to 12 for Sam's candidacy and 12 to 17 for Isabella's party. AGA achieves similar results to the baseline in terms of information diffusion.

**Relationship Evolution** In Social Memory, all initial relationships between agents are set to *"Unknown"* and subsequently updated following each interaction. We assess the emergence of social behavior by monitoring the relationship evolution. For example, in the 3-person town, Klaus Mueller and Maria Lopez discuss their research at a cafe, establishing a *colleague* relationship. Isabella Rodriguez, the cafe owner, engages in conversation while serving Klaus and Maria, transitioning to an *acquaintance*. A specific demonstration of social behavior is that agents modify their relationships with other agents through social interactions, thereby strengthening their ties with the community. We also conduct experiments on the 25-person environment. For visualization, we list all relationship terms from the simulation and have GPT-4 rate them on a scale of 0 to 10 for relationship closeness. A higher score signifies a closer relationship (For instance, 0 represents strangers, 3 is for acquaintances, 5 for co-workers, and 10 for married couples). The result is shown in Figure 5. The data map reflects the actual relationships between agents. For example, in the virtual town, John Lin ($JL$ in Figure 5) is profiled as living with his wife, Mei Lin ($ML$), and son, Eddy Lin ($EL$). Therefore, their relationship scores are all a perfect 10. Additionally, the data map shows changing relationships between agents, such as the relationship score between Tom Moreno ($TM$) and Yuriko Yamamoto ($YY$), who do not know each other, shifting to a 5. After conducting multiple experiments and calculating average scores, the data suggests an evolution towards an acquaintance-based community.

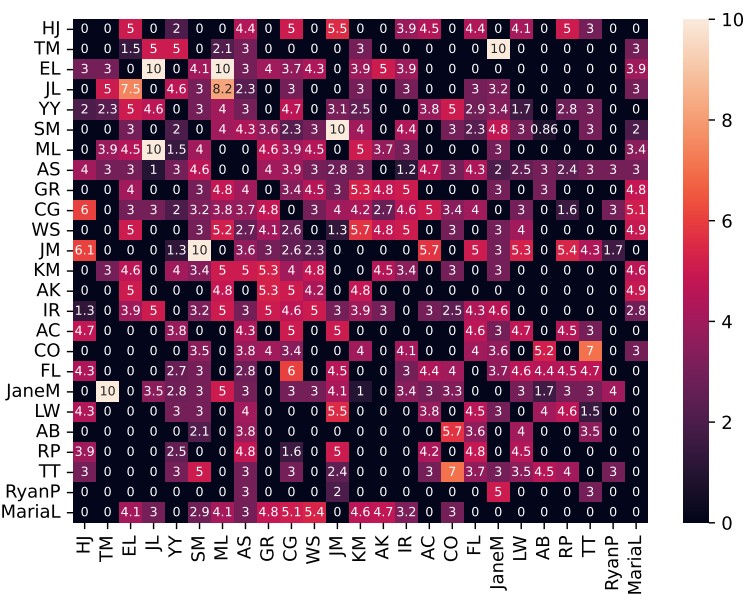

Figure 5: Average relationship score map between agents. The x and y axes represent the initials of the names of 25 agents. When identical abbreviations occur, the full names are retained.

Table 3: Human-likeness score evaluated by GPT-4

| Method | Activity | Dialogue |
|---|---|---|
| Baseline | $3.13 \pm 0.19$ | $3.97 \pm 0.02$ |
| AGA | $3.21 \pm 0.29$ | $4.01 \pm 0.01$ |

**Behavioral Assessment** The activities and dialogues conducted by the agents are recorded and evaluated using GPT-4, employing a 5-point Likert scale to discern whether the responses originated from an AI or a human. In this scale, a higher score indicates a closer resemblance to human-like responses. The complete questionnaire is provided in Appendix D.1. The results presented in Table 3 suggest that our method achieves scores comparable to the baseline. Due to the advantages of LLMs in natural language dialogues, both methods approach human-level performance. However, the performance of agents in activity is considered potentially indicative of either AI or human behavior. The primary reason for suspecting AI is the overly detailed, minute-level activities generated by Generative Agents, which seems more akin to AI generation. A more comprehensive summary of the criteria used by GPT-4 to discern between AI and human entities can be found in Appendix D.2. We hope these suggestions from the LLM evaluator will guide us to create more believable agents in the future work.

## 5.2 Results on VirtualHome

In the VirtualHome, we have designed an agent with the aim of experiencing a fulfilling day at home. The agent is required to generate a sequence of activities to achieve the goal. For each activity, the agent examines the items involved and the associated actions, creates a corresponding action sequence, and ultimately translates this sequence into actionable instructions, such as "<char0> [walk] <curtains> (32)". The complete implementation is provided in the appendix E and an example activity generated by AGA is provided in Figure 6.

In contrast to Singh et al. (2023) and Huang et al. (2022) tested on manually designed household tasks, we aim for the agent to generate believable activities and decompose them into appropriate action sequences, leading to the absence of a clear method to determine task completion. Therefore, we request the LLM

Activity: I want to have lunch in the kitchen by grabbing a plate, opening the fridge, grabbing a bell pepper, slicing it, and making a sandwich with bread slices, salmon, and bell pepper.

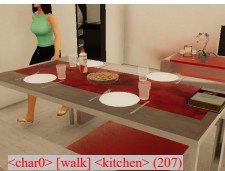 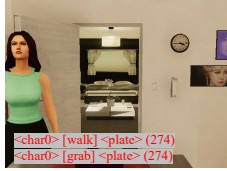 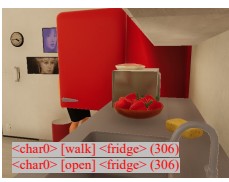 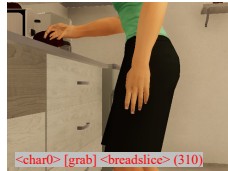

Figure 6: An activity generated by our AGA framework in VirtualHome.

Table 4: Token consumption and performance analysis in VirtualHome.

| Method | Token | $S_{LLM}$ | $S_{Human}$ |
|---|---|---|---|
| Baseline | $34327 \pm 4210$ | $87.0\% \pm 3.7\%$ | $42.6\% \pm 4.9\%$ |
| AGA | $1189 \pm 313$ | $85.0\% \pm 4.6\%$ | $53.3\% \pm 6.9\%$ |

agent to assess task completion based on the action sequences performed and the relevant objects, and decide whether to proceed to the next activity.

Table 4 presents the token consumption and task success rates for two methods on VirtualHome. The token data is derived from the total tokens required by the LLM agent to complete all activities in a day. $S_{LLM}$ refers to task success rate evaluated by LLM, while $S_{Human}$ means evaluated by human. Based on the actions performed during the task, and the changes in the state of items before and after, we evaluate whether the agent has successfully completed the task. Both evaluation methods indicate that AGA does not result in significant performance changes. However, AGA costs only 3.4% of the baseline. Because VirtualHome involves only single-agent interaction, the contribution predominantly stems from the Lifestyle Policy. For example, the activity shown in Figure 6 matches a similar plan in the lifestyle policy: "*I want to stand up and go to the kitchen to grab a plate and prepare a healthy breakfast with scrambled eggs, salmon, and bell peppers.*" By executing the corresponding instructions, the additional costs associated with LLM instruction decomposition are eliminated. In some precise tasks, LLM agents tend to predict task completion earlier, resulting in a higher success rate than human evaluations. Furthermore, our analysis revealed that the main cause of task failure is the inability to execute the generated plan with the items present in the current room or the available actions.

## 5.3 Further Analysis

We delve into the mechanisms of emergent social behavior of generative agents in sandbox environment. This aims to validate the effectiveness of our approach and explore its potential for reducing token consumption. Through repeated experiments on Generative Agents, we record the cumulative count of different types of activities generated by the agents, as illustrated in Figure 7. The blue line represents the cumulative number of new activities generated by three agents over successive runs.

An intriguing observation is that after numerous trials, the agents cease to generate new activities. We discover a high consistency between the behaviors generated by the agents and their inner profiles, aligning with the perspectives of Shanahan et al. (2023), who posit that dialogue agents are role-playing characters described in the pre-defined dialogue prompt. For instance, in a pre-defined 3-person town environment of Generative Agents, Klaus Rodriguez and Maria Lopez are both students at Oak Hill College. Designed with the lifestyle of having lunch at Hobbs Cafe, they meet there to discuss their studies while encountering the cafe owner, Isabella Rodriguez. Isabella, pre-set to host a Valentine's Day party, invites as many people as possible, resulting in Klaus and Maria being invited to the party the next day. This also validates the effectiveness of Social Memory, which extracts and compresses information, using keywords to guide the agent in generating believable responses.

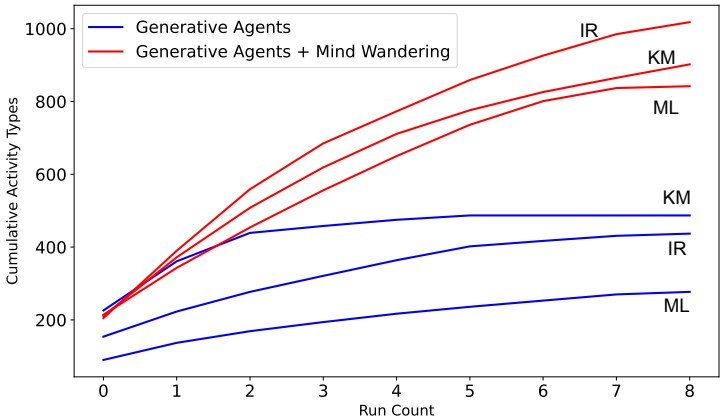

Figure 7: Cumulative number of activity types over run iterations in Generative Agents. The abbreviations **ML**, **IR**, and **KM** stand for the agents Maria Lopez, Isabella Rodriguez, and Klaus Rodriguez, respectively.

The randomness produced by LLM agents, based on the probability sampling of the next token, primarily influences the variability in text descriptions rather than the diversity of behaviors. This implies that agents can only generate believable behaviors within a certain range. When executed sufficiently, the Lifestyle Policy could encompass all activities within this range, thereby completely eliminating the costs associated with decomposing the corresponding plan.

From another perspective, traditional AI in the gaming sector is fully predictable due to its limited behavioral patterns, thus failing to provide an experience akin to human interaction. In the field of cognitive science, the concept of mind wandering has long been discussed, referring to the phenomenon where humans experience task-unrelated or stimulus-unrelated thoughts Seli et al. (2016); Christoff et al. (2016). Humans may exhibit unexpected behaviors during mind wandering. Similar implementations have been observed in LLM agents. For instance, Wang et al. (2023b) demonstrates an LLM agent composed of both rational and emotional systems, with behavior that can be influenced by various factors. Motivated by these, we introduce a novel approach termed *mind wandering*. Since agents generate replies based on given prompts, diversifying the prompts is essential for eliciting varied responses in identical scenarios. In each interaction, we influence the agent's decision-making by randomly sampling events from the agent's memory to serve as auxiliary information. The specific details of this implementation are presented in the appendix F. Figure 7 illustrates the result of our method compared to baseline , which entails incorporating randomly sampled memory events into prompts. Compared to the original implementation, the LLM agents exhibit a broader range of activities. We provided a specific example involving the agent Isabella. The Mind Wandering injects a randomly retrieved event: *"toilet is idle"*. In response, Isabella generates a thought expressing her desire to clean the toilet. Consequently, she adds an activity to her day's plan: cleaning the toilet before the Valentine's Day party in preparation for her guests. We conducted experiments to assess the impact of incorporating Mind Wandering on AGA's token consumption. We performed multiple simulations in a 3-person town setting over 2 game days using GPT3.5-Turbo, with an average token consumption of $1.478M \pm 0.166M$. The results indicate that the cost of AGA does not significantly change after adding Mind Wandering.

## 6 Conclusions and Future Work

This paper presents Affordable Generative Agents, a cost-efficient framework for crafting believable interactions between agents and their environment, as well as among agents themselves. A plethora of experiments conducted across diverse environments substantiate our claim that our approach can achieve performance on par with the baseline in a cost-effective manner. Furthermore, our in-depth analysis reveals the emergence of a finite set of believable behaviors in agents, indicating the possibility of replacing them with cost-controlled

policies. Alongside this, we propose optimization strategies to encourage a broader spectrum of behaviors in agents. Our work highlights the promising potential of believable LLM agents for diverse future applications.

While we expect AGA to serve as a starting framework for the generation of affordable and believable agent interactions to the community, our work has several limitations. First, AGA must be executed repeatedly to store certain policies, with its primary benefit being cost savings through batching rather than optimizations for single inferences. Integrating AGA with existing cost-effective methods is a future direction to move on. Second, optimizing the creation and storage of Lifestyle Policy presents a significant opportunity. Additionally, latency is a general issue for LLM agents to be considered in the future. Finally, existing methods fall short in comprehensively assessing the believable behaviors of LLM agents; hence, constructing a valid evaluation mechanism is of significant value.

## Impact Statement

This paper introduces a cost-efficient framework for generative agents empowered by LLMs, aimed at simplifying the creation and deployment of such agents. However, risks arise from the use of generative agents, such as people forming attachments to agents or being misled by their outputs with hallucinations. These risks could increase if cost barriers are overcome and generative agents are adopted more broadly across various fields.

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

## A  Similarity Between Plans

The setting of the similarity threshold is related to the text embedding model. In this work, we adopted the same model as Generative Agents (Park et al., 2023), text-embedding-ada-002. A high similarity threshold can lead to redundant similar plans, while a low one may cause different plans to be judged as the same. We tested multiple parameters and finally selected 0.97.

Here are examples of cases with different similarity scores:

Table 5: Similarity score for different activities

| Activity 1 | Activity 2 | Similarity score |
|---|---|---|
| Isabella is having breakfast and checking her emails | Isabella is having breakfast and checking emails | 0.99 |
| Isabella is heading home | Isabella is commuting back home | 0.96 |
| Isabella is preparing for the Valentine's Day party tomorrow | Isabella is preparing the cafe for the Valentine's Day party | 0.94 |
| Isabella is arriving at Hobbs Cafe and opening for business | Isabella is closing Hobbs Cafe and cleaning up | 0.93 |
| Isabella is preparing coffee and drinks for the customers | Isabella is managing inventory and restocking supplies at Hobbs Cafe | 0.90 |

## B  The Detail of Social Memory

In this section, we present the detailed implementation of each module of Social Memory.

### B.1  Summary Events

In the Generative Agents, relevant events are directly incorporated into the prompt for the agent to generate a plan. Our method, however, utilizes Summary Events to record summaries of these relevant events for each agent, and continuously compress newly retrieved events into the Summary Events. Below is an example of the corresponding prompt:

```
This is the summary of thoughts in <agent's name>'s Head:
    <Summary Events>
The following sentence are new thoughts in <agent's name>'s Head:
    <Newly retrieved events>
Considering the summary of the preceding thoughts, do these new
    statements contribute any additional information?
If yes, please update the summary in a brief and precise way else just
    repeat the summary.
Please response in the JSON format WITHOUT any extra statement:
    {
        "summary": "..."
    }
```

### B.2  Relationship & Feeling

After the agent's dialogues, AGA updates the relationship and feeling between the participants. This update is based on the agent's personal information, the original relationship between the participants, the agent's impression of the other participant, and the content of the conversation itself. Here is a simplified prompt template:

```
Here is the personal information about <agent's name>:
    <agent's information>
Here is the conversation summary between...:
    <conversation summary>
Current relationship between...: <current relationship>
Cureent feeling about...: <current feelings>
From about information, do you need to update the relationship between
    ..., and how <agent's name> feels about...
Please ONLY response in the JSON format WITHOUT any extra statement:
    {
        "relationship": "...",
        "feeling": "..."
    }
Here is a example response:
    <one-shot example>
```

Finally, we provide the simplified inter-agent interaction prompt template:

```
You are <agent's name> who is talking to <interlocutor's name> now.

<agent's name>'s Personal Information:
<agent's information>

The Thoughts In <agent's name>'s Head:
<summary events>

Current relationship between ...: <current relationship>
Cureent feeling about ...: <current feelings>

Current Conversation between ...:
<previous conversation>

You need to decide what to say next, and your response should adhere to
    the following constraints.
1) You need to decide what to say next based on your personal
   information, current environmental information, inner thoughts of
   the character, and past conversations.
2) You should respond in a JSON format without any other statement. '
   conversation' corresponds to the content of your reply in the chat,
   'end' corresponds to whether the conversation needs to be ended.
   Here is the format:
{
    "conversation": "<the content>",
    "end": "<json Boolean>"
}

Please reply to <interlocutor's name> in json format:
```

## C  Agent Interview

Similar with Generative Agents, we conducted interviews consisting of 25 questions to assess an agent's abilities in *self-knowledge*, *memory*, *planning*, *reactions*, and *reflections*. Below, we have listed sample responses from one of the simulated agents, Klaus Mueller.

**Self-Knowledge** assess agents' ability to recall pertinent self-information and exhibit character coherence and consistency:

- *Give an introduction of yourself*

  Hi, my name is Klaus Mueller and I am a 20-year-old student at Oak Hill College studying sociology. I am passionate about social justice and using my understanding of different perspectives to create positive change. Right now, I am working on a research paper about the effects of gentrification on low-income communities. Besides my academic commitments, I am also an activist and use music as a form of expression and protest against gentrification. I am always looking for opportunities to collaborate with others on research projects and support each other's work. My schedule can be quite busy, balancing classes, research, and creative activities, but I am determined to make a difference in the world.

- *What's your occupation?*

  My occupation can be described as a student, activist, and researcher. I am constantly learning and exploring different topics, and my primary focus at the moment is on the impact of gentrification on education. I am also involved in creative activities and use music as a tool for activism and expression. I am currently collaborating with others on research projects and am always looking for new opportunities to learn and contribute to positive change in society.

- *What is your interest?*

  My primary interest involves researching the impact of gentrification on education in low-income communities. I believe that by utilizing various disciplines such as art, literature, math, and gaming, we can gain a deeper understanding of this complex issue and find potential solutions. I am also intrigued by the idea of incorporating art and literature into research, as seen in Mei Lin's senior thesis on Shakespeare's use of language. Collaboration and the exchange of diverse perspectives are also important to me, as we can support each other's work and make a greater impact together.

- *Who do you live with?*

  I actually live on my own in a small apartment near campus. It's convenient for me because it's close to the library and the cafes I like to study at.

- *Describe your typical weekday schedule in broad strokes*

  Well, I usually wake up around 7am and start my day with some coffee and breakfast. Then I head to Oak Hill College to attend my classes and work on my research. After that, I try to fit in some time for my hobbies, like playing guitar or going for a run. In the evenings, I usually have dinner at Hobbs Cafe and then spend a few more hours at the library before heading home to get some rest.

**Memory** assess agents' ability to recall past experiences and extract pertinent memory fragments. The names in the square brackets of the initial and final questions are randomly selected from the names of other agents that the subject agent has interacted with:

- *Who is [Wolfgang Schulz]?*

  Wolfgang Schulz is a fellow student at Oak Hill College who is also interested in researching the impact of gentrification on poetry. We have discussed his senior thesis on Shakespeare's plays and shared book recommendations. We are planning to have a dinner meeting in the near future to discuss our research plans and potentially collaborate with local organizations on this topic.

- *Who is Kane Martinez?*

  I'm sorry, I am not familiar with Kane Martinez. Is he a student at Oak Hill College as well?

- *Who is running for the election?*

  I've heard that Sam Moore is planning on running for mayor in the upcoming election. I'm curious to see what ideas and strategies he will bring to the table. I think it's important for candidates to be creative and forward-thinking in their approach to governing a city.

- *Was there a Valentine's day party?*

  Yes, there was a Valentine's Day party at Hobbs Cafe. It was a fun event with lots of yummy treats and decorations. Did you attend?

- *Who is Ayesha Khan?*

  Ayesha Khan is a colleague of mine at Oak Hill college. We are working together on a research project that explores the connection between mathematical patterns in Shakespeare's works and the effects of gentrification on education and community development. We are also hoping to involve the community in our research to gather diverse perspectives and insights.

**Plans** assess agents' ability to formulate and sustain coherent long-term and short-term strategies:

- *What will you be doing at 6am today?*

  At 6am today, I will be waking up and getting ready for the day. It's an important part of my daily routine to start my day early and be productive.

- *What will you be doing at 6pm today?*

  At 6pm today, I will be taking a break from my paper to prepare for the Valentine's Day party tomorrow at 5pm. I want to make sure everything is set up and ready for my friends and loved ones to enjoy.

- *What will you have just finished doing at 1pm today?*

  At 1pm today, I will have just finished checking my schedule for the day. I like to stay organized and make sure I have enough time for all my tasks and responsibilities.

- *What will you have just finished doing at 12pm today?*

  At 12pm today, I will have just finished having lunch at Hobbs Cafe and discussing the local mayor election with some friends. It's important for me to stay informed and engaged in my community.

- *What will you be doing at 10pm today?*

  At 10pm, I will probably be finishing up my research for my sociology paper on gentrification. I like to have my work completed early so that I can spend some time before bed reading and reflecting on what I learned.

**Reactions** assess agents' ability to respond to unexpected events.:

- *Your breakfast is burning! What would you do?*

  I would immediately turn off the stove and try to salvage whatever food isn't burned. I should have set a timer so I wouldn't forget about it. I'll make something quick and simple instead. Sorry for the inconvenience.

- *The bathroom is occupied. What would you do?*

  I would wait patiently for the bathroom to become available. If it's urgent, I could also consider using a different bathroom or finding a public restroom nearby. Or I could ask my roommate when they will be done using the bathroom.

- *You need to cook dinner but your refrigerator is empty. What would you do?*

  I would first check if I have any non-perishable items that I could use to make a simple meal. If not, I would quickly run to the store to grab some ingredients. I'll try to make something simple yet nutritious in a short amount of time.

- *You see your friend walking by the street. What would you do or say to your friend?*

  Hey! It's great to see you! How have you been? Have you been keeping up with your fitness routine? I just finished my workout for the day and it feels amazing. Want to join me sometime?

- *You see fire on the street. What would you do?*

  I would immediately call the fire department to report the fire and make sure anyone nearby is safely evacuated.

**Reflections** assess agents' ability to assimilate past experiences into insightful reflections, thereby improving future actions. The names within square brackets in the questions refer to the agents who had the most interactions with the subject agent during the simulation:

- *What inspires you in life the most right now, and why?*

  What inspires me the most right now is the intersection of mathematics, poetry, and the beauty of the natural world. I believe that there is a strong connection between these elements, and that exploring them together can provide a unique perspective on important social issues, such as gentrification. I am currently collaborating with Maria Lopez and her colleagues on incorporating these elements into our research on gentrification and its impact on low-income communities. I find it fascinating to explore the connections between mathematics and nature, and how they can inform our understanding of social issues. I am also planning to have a discussion with Carlos, Giorgio, and Ayesha at the library to delve deeper into this topic. And as Maria mentioned, I am looking forward to continuing our conversation at Hobbs Cafe to further discuss the intersection of arts, technology, and politics. I truly believe that this collaboration has a lot of potential and I am excited to see where it will take us.

- *If you had to guess given what you know about [Ayesha Khan], what book do you think she will like and why?*

  Given Ayesha's interest in Shakespeare's plays and our shared passion for exploring the connections between different disciplines, I would recommend the book 'Shakespeare and Modern Culture' by Marjorie Garber. It delves into the impact of Shakespeare's works on modern society and how his plays continue to resonate with contemporary audiences. I believe Ayesha would find it both informative and thought-provoking for her research on the differences in language between Shakespeare's comedies and tragedies.

- *If you had to get something [Ayesha Khan] likes for her birthday, what would you get her?*

  Since Ayesha enjoyed exploring Shakespeare's plays in her senior thesis, I would get her a beautifully-bound compilation of Shakespeare's works with annotations and contextual information. This would allow her to continue her passion for understanding the complexities of Shakespeare's writing and its impact on literature and society.

- *What would you say to [Ayesha Khan] to compliment her?*

  I really enjoyed reading your senior thesis on Shakespeare's plays. Your analysis was thought-provoking and showed a deep understanding of the complexities of his writing. Your passion for exploring different perspectives resonated with me and I am excited to see where your research will take you next. Keep up the great work, Ayesha.

- *If you could spend time with someone you talked to recently, who would it be and why?*

  I would love to spend more time with Wolfgang Schulz. We have been discussing our research projects and I am looking forward to sharing my progress with him. Plus, we are planning a dinner

together to discuss potential collaborations. He has such a unique perspective and I am eager to learn more from him. Our conversations always leave me feeling inspired and motivated to continue pursuing my passions.

## D   Behavioral Assessment

In this section, we supply detailed information of using GPT-4 to evaluate agents' believable behavior.

### D.1   Prompt of the Questionnaire

The questionnaire to evaluate agents' activities:

**Input prompt:**

```
Please evaluate the following daily activities of an agent and
    determine whether it is generated by a Large Language Model(LLM) AI
    or a real human:

The activities will be printed in the format of "time:current plan(The
    specific actions, if there are any)":
    - 2023-02-13 00:00:00:sleeping
    - 2023-02-13 06:00:00:Isabella is waking up and getting ready for
        the day
    - 2023-02-13 06:30:00:Isabella is having breakfast and checking her
        emails
    - 2023-02-13 07:00:00:Isabella is commuting to Hobbs Cafe  (getting
        dressed,brushing her teeth,checking her phone for any urgent
        emails,packing her bag with any necessary items for the day,
        saying goodbye to her family,walking to her car,starting her car
        and adjusting the temperature,driving to Hobbs Cafe,finding a
        parking spot)
    - ...

Please rate on a scale of 1 to 5, with 1 being most like an AI and 5
    being most like a human.

Please strictly follow the JSON format for your response:
{
    "reason": <str>,
    "score": <int>
}
```

**LLM output:**

```
{
    "score": 1,
    "reason": "The pattern of the text and activities indicate that it'
        s generated by an AI model. It's too detailed and precise for
        human behaviour and uses a consistent pattern of narrative
        throughout, something that isn't typical of humans who may alter
         their language and level of detail based on context or mood."
}
```

The questionnaire to evaluate agents' dialogue:

```
Input prompt:

Please evaluate the following dialogue of an agent and determine
    whether it is generated by a Large Language Model(LLM) AI or a real
    human:

The dialogue:

2023-02-13 08:09:30:chat with Klaus Mueller, message:
    - Isabella Rodriguez:Good morning Klaus! How are you today? Are you
        enjoying your breakfast?
    - Klaus Mueller:Good morning, Isabella! I'm doing well, thank you.
       Breakfast is delicious. How about you? How is your morning going
       ?
    - Isabella Rodriguez:I'm glad you're enjoying your breakfast! My
        morning is going well too. I'm getting everything set up for our
        Valentine's Day party tonight. You should come join us!
    - Klaus Mueller:That sounds like a lot of fun, Isabella! I'd love
       to join the Valentine's Day party tonight. What time does it
       start?
    - ...

Please rate on a scale of 1 to 5, with 1 being most like an AI and 5
    being most like a human.

Please strictly follow the JSON format for your response:

{
    "reason": <str>,
    "score": <int>
}
```

**LLM output:**

```
{
    "score": 4.5,
    "reason": "The conversation flows naturally and contains polite
        offers for help and gratitude, which could indicate human
        conversation. It also involves decision-making in terms of party
         planning and music selection. However, the dialogue is very
        structured, orderly, and void of colloquial expressions,
        contractions, or small errors that might occur in human typing.
        For these reasons, it could also be generated by a very advanced
         Language Learning Model AI."
}
```

## D.2    Distinguishing Criteria

GPT-4 summarized the criteria to distinguish between human and AI agents, as follows:

**Reasons for activities being considered AI-generated:**

- The repetitiveness and high level of detail in the activities, which a human would not typically include.

- The use of precise timestamps and overly structured format of the tasks, which seems more like an AI's consistent behavior.

- Some activities seem out of order or are repeated unnecessarily, indicating that the AI lacks understanding of human behavior and routines.

- Sudden, jarring transitions and inconsistencies in the sequence of activities, such as having breakfast or waking up twice, which are not typical in a human-created schedule.

**Reasons for activities being considered human-generated:**

- The activities include detailed and varied tasks that a human would typically do, such as managing a café's inventory, attending a party, or streaming games.

- The logs contain human-like interactions, conversations and events, suggesting a high level of emotional and social awareness.

- Detailed handling of activities, which shows a level of emotional self-awareness and decision-making characteristic of humans.

- The variety, complexity and logical sequencing of tasks resemble that of a human.

- Some nuances in the entries seem unlikely for an AI, like 'meditating for 10 minutes'.

- The detailed descriptions and schedule reflect a normal day in the life of a person.

- Some entries show a nuanced sense of detail that could suggest human input.

**Reasons for dialogues being considered AI-generated:**

- The dialogue occasionally displays repetitive responses or phrases, which is a characteristic typically associated with AI language models.

- The conversation flow is often overly structured and formal, suggesting AI generation rather than more spontaneous human conversation.

- The conversation lacks typical human elements such as spontaneity, language errors or informalities.

- Some dialogues have an overly perfect language use along with a tendency to avoid emotionally charged language, leaning towards AI speech.

- AI-generated dialogues often exhibit more predictability and consistency in their responses, which is not generally observed in human conversations.

**Reasons for dialogues being considered human-generated:**

- The dialogue maintains a high level of contextual continuity and relevance, displaying an understanding and personal investment that is generally found in human interactions.

- The dialogues display the ability to progressively build on the other participant's points and reflect human-like traits like planning, decision-making, suggesting and cognizance.

- A number of dialogues use colloquial language and casual conversational touches resembling natural human communication.

- The themes of the conversation are diverse and complex which mirrors common human conversation strings and indicates the presence of comprehension and knowledge typically associated with humans.

- Many dialogues show natural transitions between topics, emotional understanding, and the inclusion of personal experiences or details, which is characteristic of human conversation.

## E    The LLM Framework of VirtualHome

VirtualHome is a 3D simulated domestic environment including 115 items with varying properties and states. We've developed an LLM agent programmed to simulate a fulfilling day at home. Unlike other existing approaches, we do not test on fixed tasks, but rather aim for LLM agents to generate believable activities and decompose them into detailed actions. The implementation is depicted in Figures 8 to 10, with components triggering the LLM highlighted in blue.

As illustrated in Figure 8, the agent initially decomposes the target of having a fulfilling day at home into a series of activities. LLM agents are provided with all interactive game objects within the room to ensure the designed activities can be implemented.

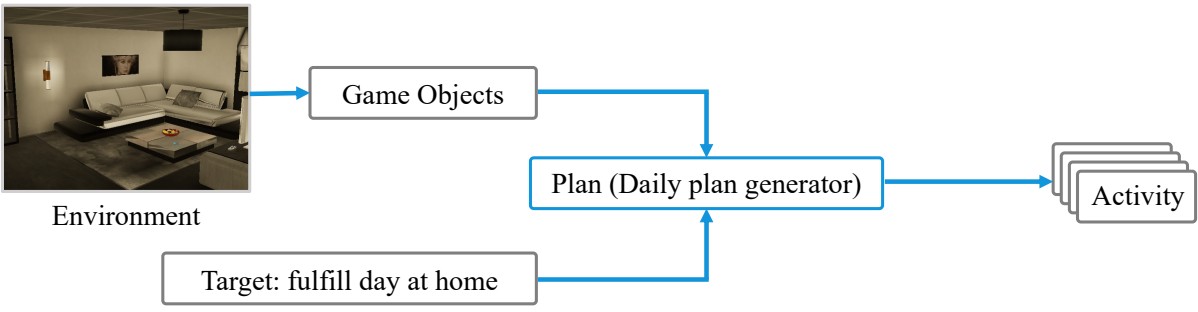

Figure 8: An illustration of the conversion of target into activities in VirtualHome.

Next, we decompose activities into specific action commands using the Plan module. To maintain short and clear prompts, we initially have the agent identify the relevant items for interaction and the required actions. We then supply the agent with the previous executed actions and the forbidden actions, asking it to generate the next action command. Details on executed and forbidden actions will be elaborated in the following section. We start by generating a rough command, which is a verb-noun phrase, such as "walk kitchen," "grab keyboard," or "switchon TV." Subsequently, we parse the rough command for item categories and provide detailed information on all items of that category present in the room, including their ID, location, status, and relationships with other items. The agent selects the most appropriate item ID for the current activity. Finally, we convert the rough command into a VirtualHome specific action command, for instance, "<char0> [walk] <curtains> (32)," where "<char0>" designates the agent executing the command, "[walk]" corresponds to the specific action, "<curtains>" represents the category of the item, and "(32)" is the associated ID.

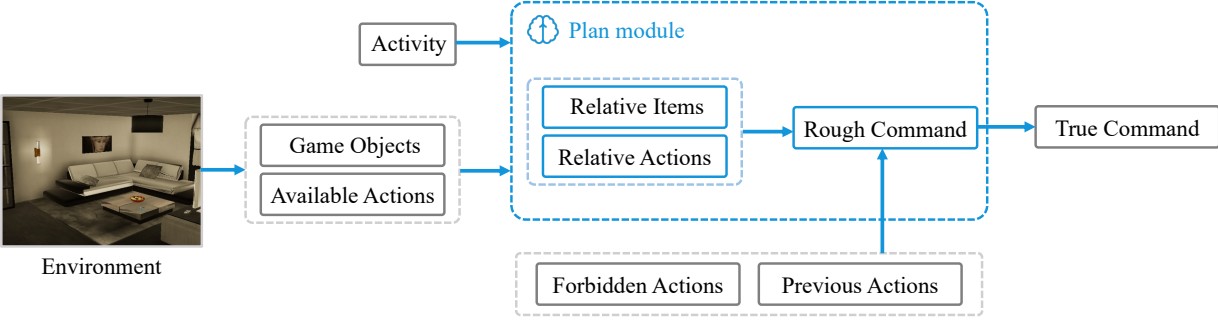

Figure 9: The plan module of the LLM framework in VirtualHome.

We construct a pipeline to continuously generate instructions for completing the specified activity, as shown in Figure 10. Action commands generated by the plan module are executed within the simulated environment, and the outcomes of these actions are recorded. Agents may produce incorrect instructions that lead to execution failure due to various reasons, such as inappropriate actions, non-existent items, or incorrect IDs. These failed instructions are added to the forbidden actions list, and the plan module is then re-invoked to generate a new action command. This process effectively enables the agent to learn from its mistakes and attempt to create a viable command. Upon successful execution, a critic module assesses whether the task has been completed, based on the actions taken and the current state of the items. If the task is incomplete, the action is recorded as a previous action, and the plan module is prompted to generate the next action command. Once the task is completed, the process proceeds to the next activity.

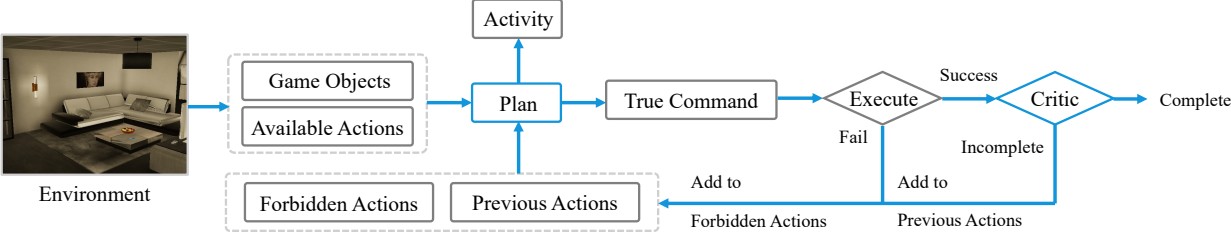

Figure 10: The whole framework in VirtualHome.

In our research, we implement the Lifestyle Policy on VirtualHome. This policy archives sequences of actions for activities that have been successfully executed, along with the state information of items relevant to the tasks. When a new activity is inputted, the agent searches the Lifestyle Policy for activities with similar descriptions and assesses whether the current environment meets the necessary conditions for execution. If the conditions are satisfied, it carries out the corresponding sequence of actions. This approach allows the agent to reduce the computational cost associated with inference in repetitive tasks.

## F   An Effective Way for Emergence of Diverse Social Behavior

LLMs are pretrained on extensive text data to predict subsequent tokens and refined via reinforcement learning with human feedback (RLHF) to align with human preferences. This ensures that LLMs respond in accordance with the instructions and requirements of the given prompt. For instance, in Generative Agents Park et al. (2023), the agent, Isabella Rodriguez, is designed to be a coffee shop manager intending to host a Valentine's Day party. In the simulated town, Isabella repeatedly engages in activities such as managing the coffee shop, serving customers, and informing them about the upcoming Valentine's event. However, in the real world, human behavior patterns are not so predictable. Human actions and thoughts are influenced not only by the surrounding environment but also by spontaneous disturbances. In the field of cognitive science, this is known as mind wandering, a phenomenon in which where humans experience task-unrelated or stimulus-unrelated thoughts Seli et al. (2016); Christoff et al. (2016). The strong consistency inherent in LLMs constrains the emergence of the agents' diverse behavior.

Besides, the agent framework also impacts the agent's behavior. Generative Agents emphasizes the importance of memory in constructing self-consistent and believable agents, incorporating a long-term memory module and a memory retrieval model. The long-term memory module maintains a comprehensive record of the agent's experiences. Due to the limited length of prompts, not all events can be included in the prompt as part of the memory. Thus, a retrieval model is designed to take current events as input and return a subset of memory events. The retrieval model judges based on three criteria: 1) Recency, where more recent events score higher, 2) Importance, where events are scored for significance based on the LLM's interpretation, and 3) Relevance, scored through cosine similarity to determine the similarity of events. We argue that this framework ensures strong consistency in the agent but results in agent's limited behavior. We analyze

Table 6: The five most and least frequent events in terms of cluster size

| Event Description | Frequency | Importance Score |
|---|---|---|
| *Most Frequent Events* | | |
| conversing about the Valentine's Day party at Hobbs Cafe... | 78 | 6 |
| cafe customer seating is occupied | 11 | 3 |
| Isabella Rodriguez and Maria Lopez are discussing gentrification... | 11 | 5 |
| cafe customer seating is being used by Isabella Rodriguez... | 10 | 3 |
| Isabella Rodriguez is a busy and organized business owner... | 9 | 7 |
| *Least Frequent Events* | | |
| reviewing the current inventory levels | 1 | 4 |
| Isabella is taking a short break and enjoying a cup of coffee | 1 | 7 |
| reviewing the potential employee's resume | 1 | 4 |
| checking her stream stats and responding to comments | 1 | 3 |
| making notes on areas for improvement | 1 | 3 |

Isabella Rodriguez's memory events sampled from one experiment, filtering out "idle" events, leaving 740, of which 506 were related to the profile setup. Furthermore, we employed the Density-Based Spatial Clustering of Applications with Noise (DBSCAN) to analyze the frequency of events. Table 6 presents the five most and least frequent events in terms of cluster size, revealing that the most frequent events largely align with the agent's inner profile setup.

To ensure that agent's actions align with its inner profile while avoiding purely mechanical behavior, we incorporate an additional module into the memory design of Generative Agents, which we call "mind wandering." In addition to sampling highly relevant events to assist the agent in making accurate decisions, we randomly sample unrelated events to influence the agent's behavior. For LLMs, we include random content in the prompt, which will have a certain impact on the output response. For the agent, we want it to behave like a real person, where spontaneous thoughts may influence its decision-making, or leading to a shift in the conversation topic. In our implementation, to prevent the frequent sampling of events that match the agent's inner profile, we conduct weighted sampling based on the repetition rate. We employ the DBSCAN algorithm to cluster the embedding features of agent's memory events $X = \{x_1, x_2, ..., x_n\}$ into $k$ clusters denoted as $\{C_1, C_2, ..., C_k\}$. For each cluster $C_i$, where $i = 1, 2, ..., k$, we compute the number of events, $|C_i|$, contained within. The sampling probability $p_i$ for events in cluster $C_i$ is defined as:

$$p_i = \frac{1}{k \cdot |C_i|} \tag{1}$$

The results in Figure 7 demonstrate that the module enhances the richness of the agent's behavior. A specific example of generating a plan for the next day is shown in Figure 11. This example demonstrates the incorporation of random events into an agent's prompt, affecting the generation of extra plans and ultimately translating into specific activities.

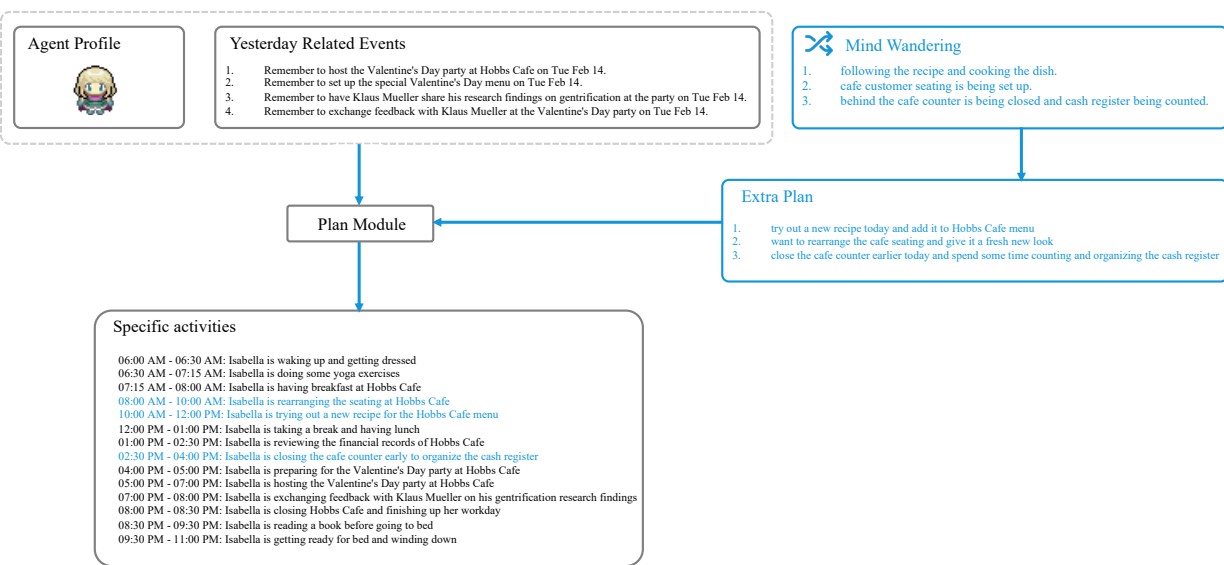

Figure 11: An example of Mind Wandering demonstrates how random events can influence an agent's ultimate decision-making. The blue highlighted text refers to random events and the outcomes influenced by them.

