# OpenReview forum: "Affordable Generative Agents"
_TMLR — Accepted by TMLR_

### Review · Reviewer_Y5qb · 2024-06-01

**Summary Of Contributions:**

Paper summary
- The paper proposes "Affordable Generative Agents" (AGA), a framework for reducing the cost of generative agents using LLMs.  Generative agents are simulated virtual agents that interact with each other and the environment.   The main idea of the work is to 1) cache and reuse responses from the LLMs and 2) reduce the context sent to LLMs.

The main contributions are:
- Two components "Lifestyle policy" and "Social Memory" for reducing the cost of generalist agents
- A set of experiments and analysis showing that the proposed components lower the token cost while still being able to come up with action plan and evolve relationships

Details:
- The AGA adds two components to the GA framework: 1) "Lifestyle policy" for agent-environment interactions, and 2) "Social Memory" for agent-agent interactions.  The "Lifestyle Policy" finds similar plans, and reuse the decomposition into subplans (thus reducing the cost of calling an LLM to decompose the plan).  The "Social Memory" summarizes the agent's interactions with other agents into relationship, feeling, and summary events, which allows for less context to be provided to the LLM.
- Experiments are conducted in two environments ("Stanford Town" and VirtualHome).  Evaluation shows that with the proposed components, AGA uses less tokens than the baseline (without the proposed components).  The quality of the agents is evaluated by using GPT-4 to score how human-like the responses are for "Stanford Town", and the task success rate is measured (by LLM and human) on VirtualHome (where agents are given a task to complete).  Experiments show that AGA is able to achieve comparable (LLM evaluation) success rate or higher (human evaluation of success rate) than the baseline.  Experiments (on "Stanford Town") also show that agents using the AGA framework has relationships that change over time with some agents having closer relationships than others. The paper also provides an analysis of when new activities stops developing, and a case study of a developed agent.

**Audience:**

Yes

**Broader Impact Concerns:**

The broader impact statement can include more specific descriptions of how the behaviors of generalist agents can mislead people (maybe undesirable behavior will be copied).

**Claims And Evidence:**

No

**Requested Changes:**

1. Writing improvement to clarify certain claims.  Some of the claims (especially in the introduction) are unsupported or unclear

    1. *"real-time, long-term invocation of LLMs to handle various types of agent interactions can result in substantial cost consumption."*
           - Is it necessarily to have real-time, long-term invocation of LLMs?

     2. *"The cost, which increases linearly with the number of interactions, is unacceptable, particularly when LLM agents are to be used as NPCs in an open world environment."* - Please provide more concrete cost estimate for an open world environment, elaborate on why the scenario is different from the two cases studied in this paper, and why the cost would be considered unacceptable.  For instance, from earlier in the introduction, the estimated cost of using LLMs for Stanford Town is just thousands of dollars and simulations of 2-3 agents is just several dollars per hours.  How is that different from the use cases considered in this paper (perhaps by connecting back to the need for "real-time, long-term invocation of LLMs")?  Why are those price points "unacceptable"? Especially considering that games can cost hundreds of millions of dollars to develop, it is not clear that thousands or numbers or several dollars per hour is "unacceptable".

    3. The sentence that follows: *"To address the issues above, there has been a significant amount of research"* implies that the cited methods are for addressing the cost of LLM agents as *"NPCs in an open world environment."*  However that is not the case.  The cited papers are work on decreasing the cost of LLM in general.  The writing is misleading as it implies that there is significant interest in the problem this paper is attempting to address.

    4. *"the main reasons these methods are unsuitable for believable behavior simulation include the following aspects"* - The logic of this paragraph is also weak as the reasons given does not appear to have much to do with the existing methods, but challenges of the problem space (lack of unified evaluation, multi-round interaction) and the observation that the authors make toward a solution to the problem (need to reduce redundancy)

    5. *"These two techniques are distilled from our insights on how a human interacts with other entities in the real world"* - Please elaborate on what are the specific insights.

2. Improvements to Figure 1
    - Figure 1 and its caption should be improved to clarify what component of the AGA framework is proposed by this work vs standard components that are part of other generalist agents.  What part calls the LLM and how the "Lifestyle policy" and "Social memory" components reduces calls to the LLM.

3. Method description should be improved to 1) provide a description of the generalist agent (GA) framework, 2) provide more details on the contributed components (some of which is found in the supplement but should be in the main paper) and how it differs from prior work.
    1.  For the "Lifestyle policy", it would be useful to include details of the embedding feature (e.g. what is being embedded, how is it embedded), how the similarity threshold is selected, discusion of how the similarity threshold will affect the tradeoff of agent quality vs cost, how the execution condition is specified / determined, etc.
    2.  For "Social Memory", it will be useful to discuss how this work choses to encode interactions the agent has with other agents vs prior work.  Also, it was hard to follow exactly what was happening with "Social Memory" -> What exactly is stored in the "Summary Events"?  How are the parts of "Social Memory" updated?  How is the dialogue prompt formed from the last round of conversation and "Social Memory"?  What exact input is provided to the LLM?

4. Improvement to discussion of experiments
    - While there is plenty of evidence that the number of tokens is reduced, there isn't enough discussion in the main paper about if / how the lowered cost (e.g. reuse of policy and condensed memory) affects agent behavior.
    - There should be a bit more comparison / discussion of how the behavior of the AGA against the baseline.
    - For instance, for the case study "Our findings indicate that our agents perform on par with the baseline, providing specific and accurate answers to the questions." => There is no comparison with the output of the original baseline, and no discussion of whether there is anything intersting to observe in the developed agent.  Similarly, there isn't any comparison for the relationship evolution, generated actity in VirtualHome, or analysis of activity types.

**Strengths And Weaknesses:**

Strengths
- The proposed components are simple and can reduce the cost of using LLMs for generalist agents
- Various experiments are conducted to show that the AGA frameworks results in lower costs

Weaknesses
- Not enough detail is provided for the two main components ("Lifestyle Policy" and "Social Memory") that are the main contribution of this work
- There is not enough comparison / discussion with the baselines in the main paper about how the proposed modules impacts agent behavior
- No other methods for lowering the cost of LLM usage is compared against
- The logic of the writing is hard to follow, with unsupported claims and missing details.

---

> ### Author Response · Authors · 2024-06-21
>
> Thank you for your thoughtful feedback. We addressed your concerns point by point.
>
> > **W1: Not enough detail is provided for the two main components...**
>
> We have added more details on the two main components in the revised version.
>
> > **W2: There is not enough comparison / discussion...**
>
> We have added more detailed comparisons and discussion with the baselines in the revised version
>
> > **W3: No other methods for lowering the cost of LLM usage is compared against**
>
> We have discussed existing methods in the paper and explained why they are not suitable for generative agents. We will emphasize this point more clearly in the revised version.
>
> > **W4: The logic of the writing is hard to follow, with unsupported claims and missing details.**
>
> We have revised the paper to present the content more clearly.
>
> > **R1: Writing improvement to clarify certain claims...**
>
>   Thank you for your feedback. We have revised the paper to ensure all claims are clearly stated.
>
> 1. **... Is it necessarily to have real-time, long-term invocation of LLMs?**
>
>    In the introduction, we define a believable LLM agent that can interact with other agents and respond to environmental feedback. In open scenarios like the Stanford Town, each interaction necessitates invoking the LLM. Over the entire simulation period, agents frequently and continuously call the LLM for various processing tasks. This refers more to long-term, high-frequency invocation of the LLM. We will clarify this in the manuscript.
>
> 2. **... it is not clear that thousands or numbers or several dollars per hour is "unacceptable".**
>
>    The costs we are discussing primarily refer to runtime costs, not development costs. In traditional game development, the bulk of expenses are indeed concentrated in the development phase, including design, programming, art, and testing. Once the game is developed, the runtime costs are relatively low, mainly involving server maintenance and content updates. However, when we introduce LLM agents as NPCs, the situation changes significantly. Each invocation of an LLM incurs a computational cost, meaning the costs increase linearly with the number of users and the duration of their interactions. This cost structure can become exponential and is unacceptable.
>
> 3. **... The writing is misleading as it implies that there is significant interest in the problem this paper is attempting to address.**
>
>    We intended to convey that there are no specific methods targeting the reduction of interaction costs for this type of LLM agents. Following the mentioned sentence, we also noted, "Nevertheless, existing methods primarily focus on specific independent tasks, solving...". We have revised the text to ensure clearer and more coherent expression.
>
> 4. **... The logic of this paragraph is also weak as the reasons given does not appear to have much to do with the existing methods...**
>
>    Thank you for your feedback. We have revised the paragraph to better align the reasons with the existing methods and address the challenges of the problem space.
>
> 5. **... Please elaborate on what are the specific insights.**
>
>    Firstly, we abstract the interaction patterns of agents into two forms: interactions with the environment and interactions with other agents. For interactions with the environment, we observe that humans tend not to overthink repetitive, mundane tasks but focus their attention when encountering novel situations. Regarding interactions with other agents, humans typically engage in various forms of interaction based on their impressions of others. Based on these insights, we propose corresponding methods utilizing LLMs. We have revised the sentence in a more rigorous way.
>
> > **R2: Improvements to Figure 1.**
>
> Thank you for your valuable feedback. In the revised version, we have redrawn Figure 1 and updated the caption.
>
> > **R3: Method description should be improved to 1) provide a description of the generalist agent (GA) framework, 2) provide more details on the contributed components...**
>
> Thank you for your valuable feedback. In the revised version, we have included a description of the GA framework and incorporated key information related to the "Lifestyle policy" and "Social Memory" into the main paper. Due to space constraints, more detailed demonstrations (such as specific prompt implementations) are provided in the appendix, but we have included clearer guidance in the 	main paper.

---

> > ### Author Response · Authors · 2024-06-21
> >
> > > **R4: Improvement to discussion of experiments**
> >
> > 1. **While there is plenty of evidence that the number of tokens is reduced, there isn't enough discussion in the main paper about...**
> >
> >    - Lifestyle Policy: In the GA framework, agents generate a plan which is then broken down into specific activites. We observed that given a plan, the LLM tends to produce homogeneous activites with only minor descriptive differences. Thus, policy reuse is nearly equivalent in this context.
> >
> >    - Social Memory: In the original GA implementation, agents' dialogues included retrieval of related events in the prompt. Our experiments showed that this often led to prompts filled with repetitive and trivial events, causing long text issues and degrading the LLM's response quality. Social memory effectively extracts and condenses important information, alleviating this problem.
> >
> >    We have elaborated on the impact of each module on agent behavior in the revised paper by presenting specific examples and implementation details compared with the baseline.
> >
> > 2. **There should be a bit more comparison / discussion of how the behavior of the AGA against the baseline.**
> >
> >    In the baseline evaluation, methods included QA interviews with agents, statistics on specific social behaviors (such as the number of people aware of Sam’s mayoral candidacy and Isabella’s party), and human evaluations. Except for human evaluations, we implemented the same methods. In the revised version, we highlighted the comparison with the original methods and offered more detailed information.
> >
> >
> > 3. **... There is no comparison with the output of the original baseline, and no discussion of whether there is anything interesting...**
> >
> >    In the revised version, we have highlighted the comparison with the baseline and added more descriptions of the agents' behaviors. In VirtualHome, household activities are primarily completed by a single agent, so there are no inter-agent relationships. We have also included an analysis of the VirtualHome agents' behaviors in the revised paper.

---

### Review · Reviewer_Ns9N · 2024-06-04

**Summary Of Contributions:**

The paper proposes the Affordable Generative Agents (AGA) to reduce the number of interactions needed in open-world scenarios. The AGA framework is based on two aspects: a Lifestyle Policy that store an arxiv of previously executed plans and retrieved them when needed and a Social Memory to store inter-agents interaction. To assess the validity of the framework, experiments were done on two environments Generative Agents and VirtualHome. In addition, the paper shows that the limits of emerging behaviors in Generative Agents.

**Audience:**

Yes

**Broader Impact Concerns:**

No concern.

**Claims And Evidence:**

Yes

**Requested Changes:**

Important requested changes:
  * More details on the metrics (e.g. 'token cost' and 'token consumption')
  * An analysis on the token cost reduction when mind wandering is applied.

**Strengths And Weaknesses:**

**Strengths**
  * Well written paper.
  * Thorough evaluations for the cost efficiency.

*Weaknesses*
  * The metrics are not properly defined. The 'token cost' and 'token consumption' are used in the figures and tables without being properly defined. What exactly is the difference between the two as in Figure 4 they seem to be used interchangeably?
  * Would it be possible that the reduced token cost is in large part due to the fact that AI agents repeat the same behaviors again and again? This would mean that the cost reduction is large because the baseline is not very great or the agents in the environments are tailored to do limited tasks (Section 5.3) but once the baseline is more human like (e.g. less repetitions) or the agents are less constrained in there conditioning the benefits of the proposed method would diminished. My remark comes from the fact that Figure 3 shows that the heaving lifting of the token cost reduction is due to the Lifestyle memory while Table 3 shows that AGA does not provide more human like activities and dialogue. In addition, Section 5.3 and Table 6 shows that the agents is heavily conditioned to repeatedly do the same things which could explain why having an arxiv of previously executed plans can largely reduce the amount of tokens needed.

---

> ### Author Response · Authors · 2024-06-21
>
> Thank you for your thoughtful feedback. We addressed your concerns point by point.
>
> > **W1: The metrics are not properly defined. The 'token cost' and 'token consumption' are used in the figures...**
>
> To avoid discrepancies due to different times and regions, we used unified LLM models and represented the cost using the number of tokens consumed (including both input and output tokens). In this context, 'token cost' and 'token consumption' refer to the same concept. We will ensure consistency in the terminology throughout the manuscript.
>
> > **W2: Would it be possible that the reduced token cost is in large part due to the fact that AI agents repeat the same behaviors...**
>
> While agents' behaviors are indeed limited, this does not imply that their actions are highly repetitive or monotonous. We simulate multiple experiments to capture a wide range of potential behaviors. The limitation arises from the fixed information provided, leading the LLM to generate responses that align with human expectations.
>
> You are correct that in a fully dynamic environment, such as a chatbot, the lifestyle policy would not be applicable. In our implementation, conversations between agents cannot be replaced, making them one of the main source of token cost.
>
> > **R1: More details on the metrics (e.g. 'token cost' and 'token consumption')**
>
> We have standardized the descriptions and added more details on the metrics in the revised version.
>
> > **R2: An analysis on the token cost reduction when mind wandering is applied.**
>
> We have included an experiment of the token cost reduction when Mind Wandering is applied in the the revised version. The results indicate that Mind Wandering has little impact on token consumption. More diverse behaviors mean more runs are needed to cover all possible plans, which has a greater impact on policy storage.
>
> | Method                  | Token Consumption       |
> |-------------------------|-------------------------|
> | Baseline                | 4.525M ± 0.364M         |
> | AGA                     | 1.417M ± 0.08M          |
> | AGA + Mind Wandering    | 1.478M ± 0.166M         |

---

### Review · Reviewer_Ng4C · 2024-06-11

**Summary Of Contributions:**

## Motivation

1. Simulated interactive agents are often costly when the interactions between humans and other agents become long
2. Current research on reducing the cost of simulating agents limits to single round QA sessions, while multi-round interactions are less supported.

The authors propose 3 challenges of *reducing the cost of believable behavior simulation of multi-round interaction*:

1. lack of unified and explicit evaluation for open-world scenarios;
2. combining multiple tasks into one single round of inference is hard due to the large amount of relevant information;
3. large-scale interaction would have redundancy that could be optimized.

## Methodology

The proposed framework, Affordable Generative Agents (AGA), introduces two key components:

1. **Lifestyle Policy**: This component minimizes costs by reusing policies for agent-environment interactions, avoiding redundant LLM invocations for repeated behaviors.
2. **Social Memory**: This component compresses auxiliary dialogue information to reduce the cost of multi-round inter-agent interactions by summarizing past interactions and relationships.

These methods aim to retain the flexible interaction capabilities of LLMs while reducing the computational and financial overhead associated with long-term, believable behavior simulations.

## Evaluation

The framework was evaluated in two primary environments:

1. **Generative Agents**: A simulated town environment with large language model agnets engage in social interactions. They tested in two environment settings: a 3-person one and a 25-person one.
2. **VirtualHome**: A single-agent 3D simulated domestic environment where agents perform household tasks through specific actions. They implemented a Generative Agent-like framework for VirtualHome.


The authors consider three different metrics: (1) token consumption, (2) relationship improvements, and (3) llm evaluation on the human-likeness score.

On  Generative Agent tasks, the authors show that AGA uses much less tokens as baselines, while maintaining similar or higher human-likeness scores as evaluated by GPT-4. In the VirtualHome environment, AGA reduced the token consumption to 3.5% while maintaining similar success rate as the baseline. With a further analysis, the authors found that Generative Agents stop to generate new activities after 2 iterations, so the authors evaluated the effect of mind wandering, which generate more diverse activities.

**Audience:**

Yes

**Broader Impact Concerns:**

Impact statement is presented.

**Claims And Evidence:**

No

**Requested Changes:**

Fix Weaknesses 1 and 2.

**Strengths And Weaknesses:**

## Strengths

1. This paper studies a timely topic: simulating social and embodied LLM agents with lower token consumptions.
2. The proposed method is effective on two different benchmarks while maintaining the performance of the baseline models.


## Weaknesses

1. This paper was not written clearly:
	1. a lot of terms undefined: *invocation* as in "real-time, long-term invocation of LLMs", *open-world* as in "open-world scenarios", *mind-wandering* introduced in Section 5 was not defined anywhere in the main text.
	2. the logics of some paragraphs and sections are weird: Paragraph 4, 1) why the lack of evaluation systems makes single-round QA unsuitable for cost-efficient model inference? I could understand 2) and 3) after reading it for several times, but they are not explained well for a reader to quickly grasp the problem of the current methods
	3. "We implement an LLM agent framework like Generative Agents on VirtualHome to validate our method." is too hand-wavy and makes the main text non-self-contained.
	4. Section 4.3 the authors mentioned 2 methods to evaluate the social behavior, but relationship evolution and llm evaluator don't seem like two parallel concepts
2. Section 5.3 seems to be an important part of evaluating AGA, but it is actually unrelated to this paper with no discussion on the token consumption. What is the main research question of this subsections?
3. To reduce the cost, using less tokens is not the only method. One can use smaller, local models or task-specific models that works as well as GPT-4. This paper only focuses on reducing the length of prompt without comparing with these alternatives.

---

> ### Author Response · Authors · 2024-06-21
>
> Thank you for your thoughtful feedback. We addressed your concerns point by point.
>
> > **W1: This paper was not written clearly:**
>
> We have revised the relevant sections of the paper to address your concerns. Regarding Section 4.3, we clarified that we aim to evaluate agents' social behaviors from two perspectives: the evolution of community group relationships and individual behavior. We have provided a revised description in the modified version.
>
> > **W2: Section 5.3 seems to be an important part of evaluating AGA, but it is actually unrelated to this paper...**
>
> In Section 5.3, we analyze the mechanism by which agents generate believable behaviors, finding that agents can only produce a limited range of activities. This finding ensures the feasibility of implementing Lifestyle Policy and implies that the policy can fully replace the inference cost associated with calling LLMs for activity decomposition. Additionally, we explore methods to enable agents to generate more diverse behaviors. In the revised version, we emphasize the connection to AGA and token consumption.
>
> > **W3: To reduce the cost, using less tokens is not the only method. One can use smaller, local models or task-specific models...**
>
> We have considered similar methods during our implementation. However, evaluating the social behavior of LLM agents poses significant challenges, particularly when different models are used. Therefore, we opted to use the same model to maintain consistency with the baseline. Additionally, our approach is designed to be independent of other cost-reduction methods and can be combined with them seamlessly.

---

### Author Response · Authors · 2024-06-21
**General Response**

Dear Reviewers,

We would like to extend our sincere gratitude to all the reviewers for their thorough and insightful feedback on our paper. Your comments and suggestions are invaluable to us and have greatly contributed to improving the quality of our work.

We are pleased to inform you that we have uploaded the revised version of our manuscript. To facilitate your review, all new content has been highlighted in **red**, and any deleted content has been **struck through in blue**.

---

### Decision · Action_Editor_gjMU · 2024-07-24

**Recommendation:** Accept with minor revision

**Comment:**

The paper proposes two ways to reduce the number of tokens processed by generative agents with some positive results on two different domains. The results show that using the methods proposed in this paper leads to less tokens and to equal or higher human-likeness in terms of behavior (note: as judged by GPT-4).

Several reviewers commented on the clarity of the writing suggesting many fixes that needed to be made. The authors were quite responsive and made significant changes to the paper. This was quite influential to the positive decision of one reviewer. Another reviewer also mentioned it. I commend the authors on being pro-active in this respect.

One of the reviewers was positive despite the methods not being especially innovative. The other reviewer was negative, specifically because the choice of domains contained exactly the properties that would lead to the methods working well. I appreciate these two criticisms and I urge the authors to consider them in future versions or in follow-up work. The first one is not a criterion that determines acceptance at TMLR, so I'm not considering it in this decision, but it could affect the long-term impact of the work. The second one is a valid criticism, but in my opinion it is not a severe limitation -- many environments of interest will have properties of repeated interaction that resemble Generative Agents and Virtual Home; in fact Q&A tasks such as technical support for companies could be another use case where there is a lot of repetition and efficient use of LLM tokens could benefit from such methods.

I am slightly disappointed by the lack of technical depth for a technical journal such as TMLR, but the evidence does support the claims and I do not think this issue is too severe that the TMLR audience would not be interested. In particular, according to the acceptance criteria, and two of three recommendations to accept, and the clear impact made by the paper, improvements to the presentation, I am happy to recommend accept.

Note: I chose "Accept with minor revision" only because the current PDF is marked up with colors and removals. The only minor revision is removing the text that is struck out and removing the colored highlights for the text that is to remain.

**Audience:**

Yes, this is of interest to the audience of TMLR. All of the reviewers answered yes to this in their official recommendation.

**Claims And Evidence:**

Yes. All of the reviewers answered yes to this in their official recommendation.